# Isolation and Characterization of Cell-Free DNA from Cerebral Organoids

**DOI:** 10.3390/ijms25105522

**Published:** 2024-05-18

**Authors:** Brian B. Silver, Ashley Brooks, Kevin Gerrish, Erik J. Tokar

**Affiliations:** 1Mechanistic Toxicology Branch, Division of Translational Toxicology, National Institute of Environmental Health Sciences, Research Triangle Park, Durham, NC 27709, USA; 2Molecular Genomics Core, Division of Intramural Research, National Institute of Environmental Health Sciences, Research Triangle Park, Durham, NC 27709, USA; gerrish@niehs.nih.gov; 3Biostatistics and Computational Biology Branch, Division of Intramural Research, National Institute of Environmental Health Sciences, Research Triangle Park, Durham, NC 27709, USA; ashley.brooks@nih.gov

**Keywords:** biomarkers, cell-free DNA, cerebral organoid, new-approach methodologies, liquid biopsy

## Abstract

Early detection of neurological conditions is critical for timely diagnosis and treatment. Identifying cellular-level changes is essential for implementing therapeutic interventions prior to symptomatic disease onset. However, monitoring brain tissue directly through biopsies is invasive and poses a high risk. Bodily fluids such as blood or cerebrospinal fluid contain information in many forms, including proteins and nucleic acids. In particular, cell-free DNA (cfDNA) has potential as a versatile neurological biomarker. Yet, our knowledge of cfDNA released by brain tissue and how cfDNA changes in response to deleterious events within the brain is incomplete. Mapping changes in cfDNA to specific cellular events is difficult in vivo, wherein many tissues contribute to circulating cfDNA. Organoids are tractable systems for examining specific changes consistently in a human background. However, few studies have investigated cfDNA released from organoids. Here, we examined cfDNA isolated from cerebral organoids. We found that cerebral organoids release quantities of cfDNA sufficient for downstream analysis with droplet-digital PCR and whole-genome sequencing. Further, gene ontology analysis of genes aligning with sequenced cfDNA fragments revealed associations with terms related to neurodevelopment and autism spectrum disorder. We conclude that cerebral organoids hold promise as tools for the discovery of cfDNA biomarkers related to neurodevelopmental and neurological disorders.

## 1. Introduction

Effective treatments for a wide range of neurological disorders, including epilepsy, neurodegenerative diseases (Dementia, Alzheimer’s, and Parkinson’s), and neurodevelopmental disorders such as autism spectrum disorder (ASD), share a common feature: establishing early and accurate diagnoses [1,2,3,4,5,6]. However, this is particularly difficult in the presence of comorbidities, as neurological symptoms can result from many underlying causes [7]. In addition, not only can many neurological disorders have roots in genetic factors, but exposure to environmental neurotoxicants can also impair cognitive health [8,9]. Effective monitoring prior to the symptomatic onset of disease is critical for combating the effects of neurotoxic exposure [10,11]. Brain tissue is sometimes obtainable for diagnostic purposes post-onset of symptomatic disease [12], but brain biopsies are highly invasive and carry a risk of potentially fatal complications [13]. Non-invasive diagnostic biomarkers of brain-related disorders and neurotoxicity are highly sought after [14,15].

Biofluids such as blood or cerebrospinal fluid (CSF) contain many potential biomarkers that hold promise as non-invasive diagnostics for early disease detection [16,17]. One such biomarker is cell-free DNA (cfDNA), which is extracellular DNA in the bloodstream or fluid surrounding cells or tissues [18]. CfDNA is believed to be largely generated through passive release by dying cells [18], but cfDNA can also be actively extruded within extracellular vesicles [19]. Characteristics of cfDNA such as concentration, fragment size, sequence, and epigenetic modifications can all potentially be indicative of disease and tissue-level events [16]. Currently, cfDNA already has several clinical applications in prenatal testing and cancer detection [20,21,22,23], but the full diagnostic potential of cfDNA has likely not yet been realized. An understanding of which cfDNA profiles are characteristic of specific tissue types and which changes in cfDNA represent deleterious events is needed to expand the diagnostic spectrum of cfDNA. Some progress has already been made in exploring the possibility of using circulating nucleic acids as potential biomarkers of brain-related disorders. For example, alterations in cfDNA have been detected in patients with cancers of the central nervous system [24], neurodegenerative diseases [25], epilepsy [26], and non-psychiatric neurological illness [27]. Differences in the nucleic acid content of extracellular vesicles have been observed in patients with Alzheimer’s Disease and are proposed to have potential as biomarkers [28]. In particular, levels of cfDNA from mitochondrial origin may be indicative of neurological and neurodegenerative disease [29].

Yet, we still lack a complete picture of which cfDNA fragments are released specifically by brain tissue. Understanding the cfDNA profiles of normal, healthy tissues is a key step towards identifying signatures of deleterious changes. Neuron-derived cfDNA has been identified in human plasma, further suggesting the potential of using circulating nucleic acids as biomarkers of neurological conditions [30]. However, identification of novel cfDNA sequences originating from brain tissues in response to brain damage or disease can be challenging in vivo due to the large proportion of cfDNA contributed from other cell and tissue sources such as white blood cells and erythrocyte progenitors [31]. In addition, the screening of toxicants to identify cfDNA signatures that are predictive, for example, of neurological damage, is generally not feasible in human subjects.

Organoids provide a tractable means of examining disease states and screening toxic compounds in a human tissue background, with extensions for the study of individualized medicine [32]. Organoids are more representative of organ tissues than 2D cell cultures or spheroids but lack the full complexity of tissues in vivo, which contain cfDNA from numerous tissue sources [31]. This permits a more isolated view of cell types and may aid the identification of biomarkers specific to certain tissues and organs. For these reasons, we chose to examine cfDNA in organoid models. Successful isolation of cfDNA from pancreatic [33], cardiac [34], lung, and gastric organoids [35] has been documented previously. However, outside of these few studies, whether cfDNA is released from other organoid types remains unexplored.

Here, we isolated and characterized cfDNA from cerebral organoids. Our analyses revealed that cerebral organoids release cfDNA in quantities sufficient for whole-genome sequencing (WGS), droplet-digital PCR (ddPCR), and other downstream assays. Intriguingly, we observed that the recovered cfDNA contained many sequences that were aligned with genes associated with synaptic development and neurodevelopmental disorders. Overall, our study demonstrates the suitability of cerebral organoid systems as a tool for the study of cfDNA. Ultimately, this approach could be used to uncover potential novel cfDNA biomarkers associated with neurodevelopment, neurological disease, and neurotoxicity. 

## 2. Results

### 2.1. Cerebral Organoids Generated for cfDNA Collection Express Markers of Neural Differentiation

Cerebral organoids were generated by seeding and differentiating H9 human embryonic stem cells (hESCs) into embryoid bodies, which were then embedded in Matrigel and guided towards neural lineage (Figure 1A) using a commercially available kit (see Section 4). Immunofluorescence staining of mature organoids revealed cells with colocalization of the neural markers β3-Tubulin and Map2, as well as populations of cells expressing the neural stem cell progenitor marker Nestin (Figure 1B,C). Consistently, we observed expression of proteins associated with neural differentiation as the organoids matured (Figure 1D). Conditioned media samples were collected from the organoids at several time points during their growth for cfDNA extraction (Figure 1E).

### 2.2. Cerebral Organoids Release cfDNA of Both Mitochondrial and Genomic Origin

We observed that the total cfDNA concentration increased between growth day 31 and maturation (day 41) of the organoids (Figure 2A). To account for changes in cell number during organoid growth, we normalized the concentrations of cfDNA to levels of genomic DNA (gDNA) collected on the same growth days. The normalized cfDNA concentration was highest on growth day 10 (Figure 2B), indicating that the cfDNA output per cell may be higher early in the organoid growth process, a finding we previously observed in cardiac organoids [34]. However, as discussed in Section 2.3 below, it should be noted that we also identified a high concentration of contaminative non-human DNA in the day-10 cfDNA samples (see Appendix A), preventing an accurate comparison of these time points. The recovered cfDNA from mature cerebral organoids consisted primarily of small fragments (100–200 base pairs) (Figure 2C).

Both mitochondrial and nuclear DNA can contribute to total cfDNA concentration, and the contribution of DNA from each of these sources can be used to further characterize cfDNA [19]. To assess the abundance of each of these potential sources of cfDNA in our samples, we employed droplet digital PCR (ddPCR) using probes targeting regions within mitochondrial or nuclear-localized genes. We observed that sequences corresponding to cfDNA of mitochondrial origin (mt-cfDNA) were detectable in cfDNA taken from cerebral organoids at several time points during growth (Figure 2D). Likewise, we were able to detect cfDNA sequences corresponding to gene regions within the nuclear-localized genes β3-tubulin, nestin, NFATC1, and OCT 3/4 in cfDNA released from mature cerebral organoids (Figure 2E). These data demonstrate that cerebral organoids release cfDNA in quantities sufficient for the detection of specific sequences using ddPCR.

### 2.3. Whole-Genome Sequencing of Cerebral-Organoid-Derived cfDNA Reveals Novel, Consistently Detectible Regions

To gain a more comprehensive understanding of the makeup of cfDNA derived from cerebral organoids, we performed whole-genome sequencing (WGS) of cfDNA isolated on growth days 10, 21, and 41. Surprisingly, this analysis revealed a high fraction of DNA of rodent origin in cfDNA taken from media conditioned by cerebral organoids on growth day 10. This non-human DNA fraction decreased on day 21 and nearly disappeared by day 41, leaving cfDNA of almost entirely human origin (Appendix A). We suspect that this contaminative rodent DNA in the early and mid- time points originated from the Matrigel in which the organoids were embedded on growth day 7. We therefore decided to focus our subsequent analyses on cfDNA recovered from mature organoids (growth day 41).

WGS showed that the majority of cfDNA fragments isolated from the day 41 cerebral organoids were 100–200 bp long (Figure 3A), consistent with our previous fragment analysis data. Sequences that aligned to the mitochondrial genome showed a higher density of shorter fragments (<100 bp) compared to sequences aligning to autosomal regions (Figure 3B). To confirm the presence of sequenced regions in our cfDNA samples, we designed custom ddPCR probes targeting a selection of cfDNA sequences from a range of FPKM values located across different genes. We were able to consistently detect these sequences in cfDNA from mature cerebral organoids (Figure 3C). This indicates that novel sequenced reads are present and detectable in cerebral organoid-derived cfDNA using ddPCR.

Genome-wide peak calling was performed to identify regions with an increased signal in each of the biological replicates. The union of intersecting peaks was classified as regions of interest (ROIs) for downstream analyses (Appendix A). A total of 3406 ROIs were identified, ranging from 169 to 6314 bp in length. To examine the distribution of ROIs across the genome, the ratio of peak counts to chromosome length (e.g., number of peaks per 100 kb length) was calculated for each chromosome (Figure 4). The majority of peak ratios were <0.21, with a notable exception for chromosome 21, with a ratio of 0.24 (z-score = 2.55).

### 2.4. Cerebral Organoids Release cfDNA Corresponding to Gene Regions Associated with Brain Development and Neurological Disorders

We found that 2282 unique genes intersected the ROIs. Functional enrichment analysis of these genes, including with regard to Gene Ontology (GO), Disease Ontology (DO), KEGG, and REACTOME, revealed terms related to brain function and development (Figure 5). Specifically, terms related to neuron and synapse formation were among the most prominent Cellular Component-associated GO terms. Synapse organization, regulation of nervous system development, and axonogenesis were highly represented in the Biological-Process-associated GO terms. Neuronal system was the most prominent pathway term identified using the REACTOME database [36]. Additionally, calcium signaling and axon guidance pathways were identified using the KEGG database. Intriguingly, Disease Ontology terms associated with intellectual disability and autism spectrum disorder were also identified. Together, these data suggest that cfDNA released from cerebral organoids maps to gene regions associated with synaptic and neuronal development as well as neurodevelopmental disorders. This observation supports our hypothesis that cerebral organoids can be used to study potential cfDNA biomarkers relevant to neurological development and disorders in a human tissue background. 

### 2.5. Repetitive Elements May Influence Fragmentation of Cerebral-Organoid-Derived cfDNA

Repetitive elements, particularly retrotransposons and satellite repeats [37,38,39], have previously been reported to be implicated in neurological disease. We were therefore interested in exploring the fragmentation patterns of cfDNA associated with repetitive elements in our cerebral-organoid-derived dataset. Notably, we found that ROIs intersecting retrotransposons or acro satellite repeats showed a higher density of longer fragments (Figure 6). Together, from these data, we conclude that cfDNA recovered from mature cerebral organoids shows non-random patterns of fragmentation and contains features of relevance to neurological development and disease. 

### 2.6. Patterns of Differential Methylation May Be Observable in Cerebral-Organoid-Derived cfDNA

Changes in gene expression during differentiation or upon neurotoxicant exposure may alter the abundance of sequences present in cfDNA through changes in histone wrapping, protein binding, or other epigenetic modifications that impact chromatin accessibility [40]. Epigenetic modifications such as methylation are known to alter the composition of cfDNA [41,42,43]. Therefore, we wished to further explore the cfDNA sequences in our dataset that might overlap with regions associated with epigenetic modifications. Previously, it was observed that differentiation of hESCs to cerebral organoids changed patterns of methylation [44]. These differentially methylated regions (DMRs) were defined by comparing mature cerebral organoids to undifferentiated hESCs with an identical genetic background. We intersected our ROIs with the reported DMRs from this previously published dataset [44]. We found that our ROIs overlapped with hypomethylated regions significantly more often than with hypermethylated regions (*p*-value = 4.514 × 10^−12^).

## 3. Discussion

CfDNA has been isolated successfully from a few organoid systems [33,34,35]. Yet, whether additional organoid models, such as brain organoids, release cfDNA in measurable quantities remains an open question. We have shown here that cfDNA can be consistently recovered from cerebral organoids in quantities sufficient for downstream analyses, including WGS and ddPCR. Further, we have demonstrated that WGS is a tractable method for gaining a broad understanding of the overall composition and patterns associated with cfDNA release from cerebral organoids. Our observation that cfDNA released from mature cerebral organoids was enriched in gene regions corresponding to GO terms such as “neuronal cell body”, “synaptic membrane”, “synapse organization”, “neuronal system”, and “axon guidance” was particularly intriguing and suggested that cerebral organoids release cfDNA that is reflective of gene regions relevant to brain tissue. This methodology could be extended to additional organoid types and potentially used to identify biomarkers associated with exposure to toxins or disease states in specific cell types or tissues. Increased presence of cfDNA sequences intersecting hypomethylated regions as we observed is suggestive of open, accessible chromatin, which could be associated with altered transcriptional activity. Although speculative, this analysis suggests that cfDNA derived from cerebral organoids contains methylation signatures relevant to differentiated cerebral tissue. Future studies might aim to directly examine methylation markers in cerebral-organoid-derived cfDNA.

The abundance of specific cfDNA sequences in a biofluid sample could potentially serve as a biomarker of tissue-level changes or disease. Recently, cfDNA of neuronal origin has been found to be elevated in the blood plasma of patients with Alzheimer’s disease [30]. In addition, mt-cfDNA levels in CSF have been observed to fluctuate in persons with neurodegenerative disease. Specifically, decreased mt-cfDNA levels were observed in patients with Parkinson’s disease, while increased mt-cfDNA levels were correlated with dementia [25]. Notably, we were able to detect mt-cfDNA in media conditioned by cerebral organoids using ddPCR. This suggests that cerebral organoids indeed release cfDNA sequences that are relevant to human disease. Also, we found that specific sequences of cfDNA derived from cerebral organoids are consistently detectable with ddPCR. This demonstrates the potential of cerebral organoids as tools for the development of ddPCR assays to detect novel cfDNA sequences. Previously, ddPCR was successfully used to identify the presence of brain somatic mutations in CSF-derived cfDNA and proposed as an alternative to directly analyzing brain tissue, which is difficult to obtain [26]. However, predicting exactly which sequences will be present and detectable at the cfDNA level in human biofluids is difficult, as many mechanisms could contribute to cfDNA release [45]. Further, cfDNA can originate from many cell types in vivo [31], making it difficult to determine which cfDNA features or sequences may be especially relevant to a specific organ and reflective of tissue damage or disease. Organoid systems may provide a tractable means of identifying cfDNA sequences relevant to specific tissue types, toxicant exposure, or disease models that could then be evaluated in a more targeted manner in human biofluids.

In addition to the potential of cfDNA to serve as a diagnostic biomarker of neurodegenerative disease, cfDNA may have use in the identification of neurodevelopmental disorders. For example, early and accurate diagnosis of conditions such as ASD is essential for managing symptoms and improving function and quality of life in adulthood. However, the diagnosis of ASD is often difficult, especially in the presence of comorbidities, due to the overlap of symptoms with other cognitive disorders [46]. In addition to advancements in cognitive and behavioral testing, there is a growing need for the identification of biomarkers of ASD [47,48]. We observed that cerebral organoids release cfDNA sequences aligning to gene regions associated with Disease Ontology terms related to ASD and intellectual disability. This finding was quite intriguing as it suggests that cerebral organoids might be used to identify features of cfDNA that could be investigated as potential biomarkers of ASD or other neurodevelopmental disorders.

In summary, here, we evaluated the ability of cerebral organoids to release cfDNA in quantities sufficient for analyses, including WGS and ddPCR. This strategy could be further applied to investigate potential cfDNA targets in cerebral organoid models of disease, neurodevelopmental disability, or neurotoxicant exposure. Future research might seek to investigate the presence of cerebral-organoid-derived cfDNA sequences in human biofluids. Overall, this study demonstrates the utility of cerebral organoid models as a tool for the identification of potential cfDNA targets relevant to brain tissue. Increasing our knowledge of non-invasive biomarkers such as cfDNA could ultimately help develop strategies for detecting and treating brain-related disease and disorders earlier.

### Study Limitations

The sequencing of cfDNA isolated from cerebral organoids on different growth days revealed a high degree of DNA of rodent origin early in the development of the organoids. All sequences of non-human origin aligned to the mouse genome, and some shared homology with the rat genome (Appendix A). Therefore, we suspect that the source of the contaminative DNA was the Matrigel scaffold in which the organoids were embedded on growth day 7. Consequently, care must be taken in interpreting differences in cfDNA concentration or sequence abundance between early and late time points during organoid maturation. For these reasons, we chose to focus our sequencing analyses on mature cerebral organoids (day 41) after the deterioration of the Matrigel scaffold and the disappearance of contamination by non-human DNA. Further research is needed to definitively determine whether quantitative differences exist between cfDNA sequences released between early and late time points of cerebral organoid growth. Although they are beyond the scope of this study, such efforts might explore Matrigel-free systems of culturing or methods of isolating human cfDNA sequences from potential contaminants present in cell culture systems. As Matrigel is widely used in culture models throughout the scientific community, further analysis of DNA present in Matrigel is an important topic for continuing investigation. Towards this end, examining whether Matrigel is unequivocally a source of rodent DNA contamination through sequencing or other assays would be a valuable area of future study. In addition, further investigation of cfDNA in cerebral organoid models, for example, through sequencing at earlier time points, may provide deeper insight into how contaminative rodent DNA impacts culture models.

This study examined only cfDNA and not cell-free RNA or other molecules released by cerebral organoids. An intriguing area of future study would be to investigate cell-free RNA released by the cerebral organoids, as a potential biomarker of transcriptional processes. Presumably, regions of the genome related to cerebral differentiation are accessible for active transcription (euchromatic). These open regions appeared to be overrepresented in the cfDNA, possibly due to altered accessibility to DNases or differences in rates of degradation and release into the extracellular medium. Although a full investigation of methylation markers and their influence on cfDNA release is beyond the scope of this study, future work in this area could yield valuable insight into the mechanisms of cfDNA release in cerebral tissues.

## 4. Materials and Methods

### 4.1. hESC Culture

H9 human embryonic stem cells (hESCs) were maintained in mTeSR+ medium (Thermo Fisher, Waltham, MA, USA) at 37 °C in an atmosphere with 5% CO_2_. Cells were passaged between 60 and 80% confluence, using 0.5 mM of EDTA/PBS (Gibco, Grand Island, NY, USA) to dissociate cells. Plates were pre-coated with a concentration of 1.2 µL/mL/cm^2^ Matrigel (Corning, Corning, NY, USA) for 30 min at 37 °C in DMEM/F:12 (Thermo Fisher). Cultures were maintained for a maximum of 10 passages and regularly examined for morphological changes indicative of differentiation.

### 4.2. Cerebral Organoid Generation

H9 ESCs were used to generate cerebral organoids with the StemDiff Cerebral Organoid Kit (StemCell Technologies, Vancouver, BC, Canada), following the manufacturer’s instructions. Briefly, cells were trypsinized with 1× TrypLE (Gibco)/0.5 mM of EDTA and resuspended in EB Formation Medium (StemDiff Cerebral Differentiation Kit; StemCell Technologies) supplemented with 10 µM of Y27632 (Tocris, Bristol, UK). Cells were seeded at a density of 2 × 10^4^ cells/mL, with 50 µL per well, on a 384 round-bottom ultra-low attachment plate (Corning). The plate was centrifuged at 200× *g* for 5 min to coalesce the cells. This was defined as day 0 of growth. On growth days 1 and 3, 40 µL of media was replaced with EB Formation Medium (StemDiff Cerebral Differentiation Kit; StemCell Technologies) and incubated for 48 h at 37 °C. On growth day 5, 40 µL of media was replaced with induction medium (StemDiff Cerebral Differentiation Kit; StemCell Technologies), and the plate was incubated again for 48 h at 37 °C. On growth day 7, the tissues were rinsed 1× with 30 µL Phosphate-Buffered Saline (PBS; Thermo Fisher) and individually embedded in 15 µL of Matrigel (Corning) on an embedding sheet (StemCell Technologies). After 30 min incubation at 37 °C to solidify the Matrigel, the tissues were transferred to a 24-well ultra-low attachment plate (Thermo Fisher) in 1 mL of Expansion Medium (StemDiff Cerebral Differentiation Kit; StemCell Technologies) at a density of four tissues/well and incubated at 37 °C on an orbital shaker for 72 h. On growth day 10, the medium was changed to 1 mL of Maturation Medium (StemDiff Cerebral Differentiation Kit; StemCell Technologies), and the media was changed every 3–4 days until maturity (day 41).

### 4.3. Nucleic Acid Extraction and Quantification of Gene Expression

Genomic DNA (gDNA) from organoids and cfDNA from conditioned media were extracted using the Promega Maxwell^®^ RSC instrument (Promega, Madison, WI, USA) with the Tissue DNA Kit or ccfDNA Plasma Kit, respectively. DNA concentrations were measured using the Promega QuantiFluor^®^ ONE dsDNA System. For gDNA extraction, 3–4 organoids were homogenized according to the manufacturer’s protocol. CfDNA was isolated from 1 mL of medium conditioned by 4 organoids (2–3 technical replicates per batch, with a total of 3 independent batches). Prior to cfDNA extraction, all conditioned media were centrifuged for 10 min at 1600× *g*, the top 800 µL supernatant was centrifuged for 10 min at 16,000× *g*, and the top 400 µL was collected and stored at −80 °C until extraction. Prior to extraction, conditioned media was thawed at 4 °C to prevent degradation. Gene expression and copy number were evaluated using droplet digital PCR (ddPCR; Bio-Rad, Hercules, CA, USA), using either 0.5 or 1 ng of cfDNA per reaction.

### 4.4. Protein Detection and Quantification

In preparation for Western Blot analysis, organoid tissues were collected at several timepoints during growth (days 10, 13, 17, 31, and 41), rinsed 1× with PBS, and homogenized in RIPA buffer (Thermo Fisher Scientific, Waltham, MA, USA) supplemented with 25× Complete Protease Inhibitor (Thermo Fisher Scientific). Samples were centrifuged for 15 min at 14,000× *g* at 4 °C. The supernatant was removed and stored at −80 °C until analysis. Protein concentration was calculated using the Pierce^TM^ BCA assay (Thermo Fisher Scientific). Protein (3.5 µg) was incubated for 10 min at 70 °C with NuPage Reaction Buffer (Thermo Fisher Scientific) and NuPage Reducing Agent (Thermo Fisher Scientific) prior to being loaded onto NuPage gel (4 to 12%, Bis-Tris; Thermo Fisher Scientific). The gel was run for 35 min at 200 V in NuPage MES Run Buffer (Thermo Fisher Scientific), to which NuPage Antioxidant (Thermo Fisher Scientific) was added. After migrating through the gel, the proteins were transferred to nitrocellulose membranes using an iBlot™ Gel Transfer Device (Thermo Fisher Scientific). Ponceau S solution (Sigma, St. Louis, MO, USA) was used to confirm protein transfer to the membrane. Membranes were then blocked for 15 min in EveryBlot Blocking Reagent (Bio-Rad, Hercules, CA, USA) and incubated in primary antibody solution overnight at 4 °C. The following antibodies were diluted 1:1000 in Bio-Rad EveryBlot Blocking Reagent: rabbit anti-GAPDH (Bio-Rad), rabbit anti-PSD95 (Cell Signaling, Danvers, MA, USA), rabbit anti-synaptophysin (Cell Signaling), and rabbit anti-Gap43 (Cell Signaling). Membranes were washed 3 × 5 min in 1x tris-buffered saline (TBS; Bio-Rad), to which 0.1% Tween-20 (Sigma) (TBST) was added. Membranes were then incubated for 45 min at room temperature in secondary antibody solution: anti-rabbit-HRP (Novus, St. Charles, MO, USA) diluted 1:5000 in EveryBlot Blocking Reagent (Bio-Rad). Membranes were washed 3 × 15 min in TBST. HRP was visualized using SuperSignal™ West Pico Plus Chemiluminescent Substrate (Thermo Fisher Scientific) according to the manufacturer’s protocol.

### 4.5. Immunofluorescent Staining

Mature cerebral organoids were fixed, cleared, and immunostained following a published protocol [49], as previously described [34]. Primary antibodies were diluted 1:200 in blocking solution (3% Bovine Serum Albumin/PBS; Sigma): chick anti-β3 tubulin (Cell Signaling), mouse IgG1 anti-nestin (Cell Signaling), and rabbit anti-map2 (Cell Signaling). Secondary antibodies were diluted 1:200 in blocking solution: Alexa 647 donkey-anti-rabbit (Invitrogen, Waltham, MA, USA), Alexa 488 goat-anti-mouse (Invitrogen), Dylight 550 goat-anti-chicken (Thermo Fisher Scientific), and DAPI (1 µg/mL; Sigma). Tissues were imaged on 96-well optical plates (Thermo Fisher Scientific, M33089) using a Zeiss LSM 880 inverted confocal microscope with AiryScan (Jena, Germany).

### 4.6. Sequencing of cfDNA Fragments from Cerebral Organoids

CfDNA collected from media conditioned by cerebral organoids on growth days 10, 21, or 41 (with three biological replicates each) was used to prepare sequencing libraries using the ThruPLEX Tag-seq Kit produced by Takara Bio in accordance with the manufacturer’s protocol. Libraries were sequenced on an Illumina NovaSeq 6000 platform to generate 150-base-pair (bp) paired-end reads (Illumina, San Diego, CA, USA).

### 4.7. Bioinformatic Analyses

The Takara ThurPLEX Tag Seq HV processing pipeline for UMIs was applied for data processing (https://www.takarabio.com/learning-centers/next-generation-sequencing/dna-seq-protocols/using-umis-with-thruplex-tag-seq-hv, accessed on 1 April 2023). In summary, adapter sequences were removed from raw fastq files with Trimmomatic v.0.39 [50]. Unique molecular identifiers (UMIs) were appended to an unmapped BAM RX tag using the fgbio v.2.1.0 FastqToBam function and subsequently removed from the trimmed reads with picard v.2.26.9 SamToFastq (https://fulcrumgenomics.github.io/fgbio/; https://broadinstitute.github.io/picard/, accessed on 1 April 2023). Processed reads were aligned to the Hg38 genome with Bowtie2 v.2.5.2 and converted to sorted BAM format with picard SortSam and samtools v.1.18 view functions [51,52]. The picard MergeBamAlignment function was used to incorporate the UMI information from the unmapped BAM file to the sorted, aligned BAM. Filtered, deduplicated consensus reads were obtained in unmapped BAM format with the fgbio functions CorrectUmis, GroupReadsByUmis, and CallMolecularConsensusReads. Consensus reads were extracted with SamToFastq and aligned with Bowtie2. The alignments were sorted and converted to BAM format with SortSam and the samtools view module. The UMIs present in the unmapped BAM files containing consensus reads were incorporated into the Bowtie2 alignments to generate the final BAM files used for all downstream processes. Specific flag setting for all pipeline steps followed recommendations from the Takara pipeline (https://www.takarabio.com/learning-centers/next-generation-sequencing/dna-seq-protocols/using-umis-with-thruplex-tag-seq-hv, accessed on 1 April 2023). Potential contamination of cfDNA with rodent DNA was assessed with the command line utility fastqscreen v.0.15.2 using the fastq files containing consensus reads [53].

Fragment length analysis was performed to compare differences in fragment length density between autosomal, mitochondrial, or genomic repeat classes. The day 41 cfDNA replicates were merged into a single BAM file using the samtools merge function. Estimated fragment lengths were extracted from the 5’ read of properly paired reads in the merged BAM file and plotted for relevant comparisons. The Repbase RepeatMasker library 20140231 was queried to determine reads mapping to Acro1 satellite or SVA retrotransposon regions in the genome [54].

The macs2 v.2.2.9.1 peak caller with the flags -f BAMPE --nomodel -p.01 -g 2652783500 --broad --broad-cutoff 0.01 was employed to identify regions of signal enrichment in each of the Day 41 replicates [55]. The union of broad peak intersections between the replicates comprised the final regions of interest (ROIs). Gencode v.32 Hg38 genes that intersected with ROIs were subject to pathway enrichment with clusterProfiler v.4.8.2 in the R v.4.3.1 programming environment [36]. Genomic coordinates for hypo- and hypermethylated regions in day 40 organoids were retrieved from Luo et al. (2016) [44]. The Hg19 methylation coordinates were increased from Hg19 to Hg38 with the UCSC liftOver command line utility [56]. Hypo- and hypermethylated regions were intersected with ROIs, and Fisher’s exact test was applied to determine the significance of overlaps. All interval intersections and merges were performed with the bedtools v.2.25.0 intersect and merge functions unless otherwise stated [57].

## Figures and Tables

**Figure 1 ijms-25-05522-f001:**
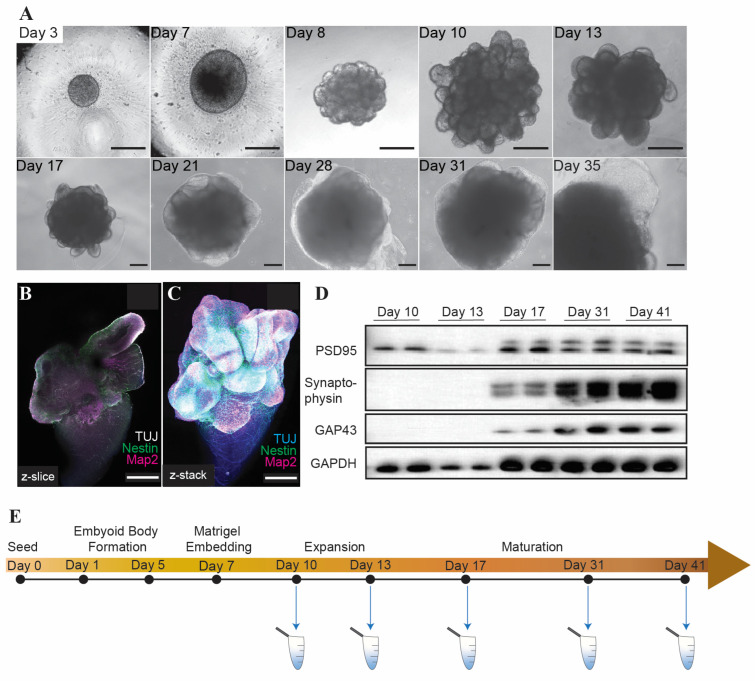
Cerebral organoids derived from H9 ESCs express neural differentiation markers. (**A**) Brightfield images of cerebral organoids during growth and differentiation. Scale bars represent 100 µm. Immunofluorescence z slice (**B**) and z-stack (**C**) showing β3-tubulin (TUJ), Nestin, and Map2 in mature (day 41) cerebral organoids. Scale bars represent 500 µm. (**D**) Western blot showing expression of PSD95, Synaptophysin, GAP43, and GAPDH (loading control) in cerebral organoids during maturation. (**E**) Schematic illustrating timepoints of cfDNA collection in cerebral organoids.

**Figure 2 ijms-25-05522-f002:**
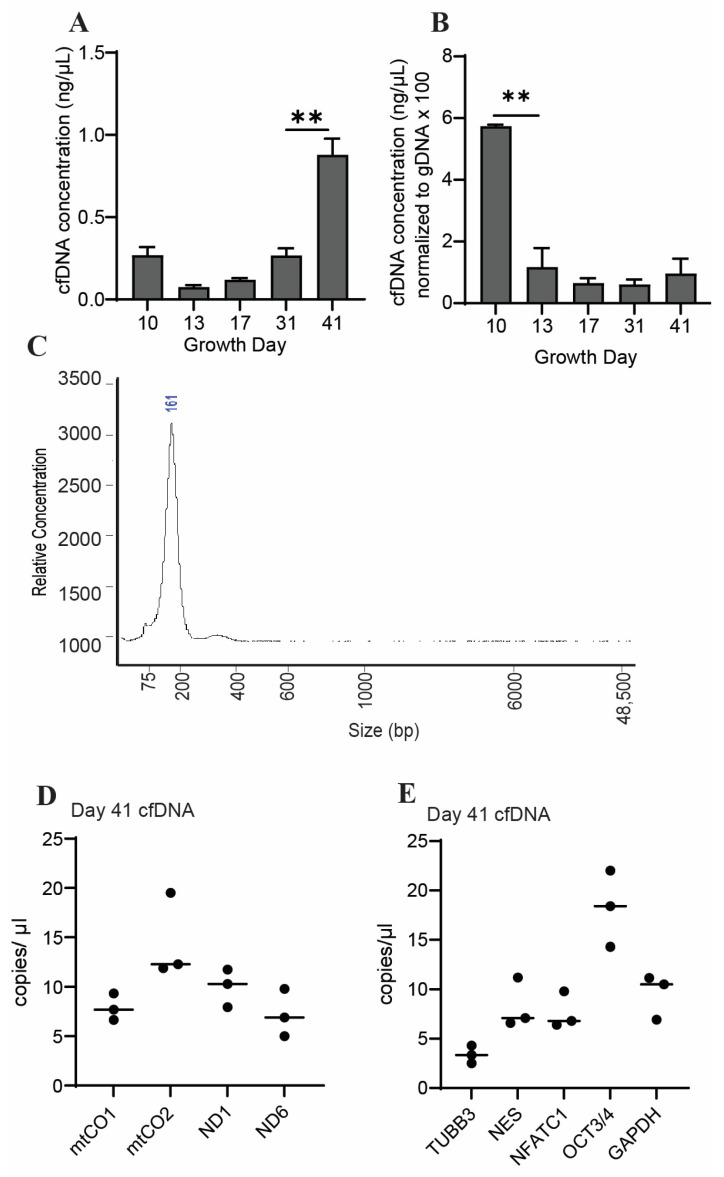
CfDNA is recoverable from cerebral organoids in quantities sufficient for downstream analysis. (**A**) Shown are cfDNA concentrations taken from n = 3 biological replicates during cerebral organoid differentiation. (**B**) Cerebral-organoid-derived cfDNA concentrations from panel (**B**), which were normalized to gDNA concentrations (multiplied by 100 to facilitate axis readability). Graphs show average concentrations of n = 3 biological replicates +S.D. Samples were compared using an unpaired *t*-test with Welch’s correction. **, *p* < 0.01. (**C**) Representative electropherogram showing fragment sizes of cfDNA derived from mature cerebral organoids. (**D**) Abundance of cfDNA sequences detected using ddPCR with probes corresponding to the following mitochondrial genes: mtCO1, mtCO2, ND1, and ND6. (**E**) Abundance of cfDNA sequences recovered from mature cerebral organoids detected using ddPCR with probes corresponding to sequences within the genes β3-tubulin (TUBB3), nestin (NES), NFATC1, or Oct 3/4 (OCT). Graphs show individual data points and associated means.

**Figure 3 ijms-25-05522-f003:**
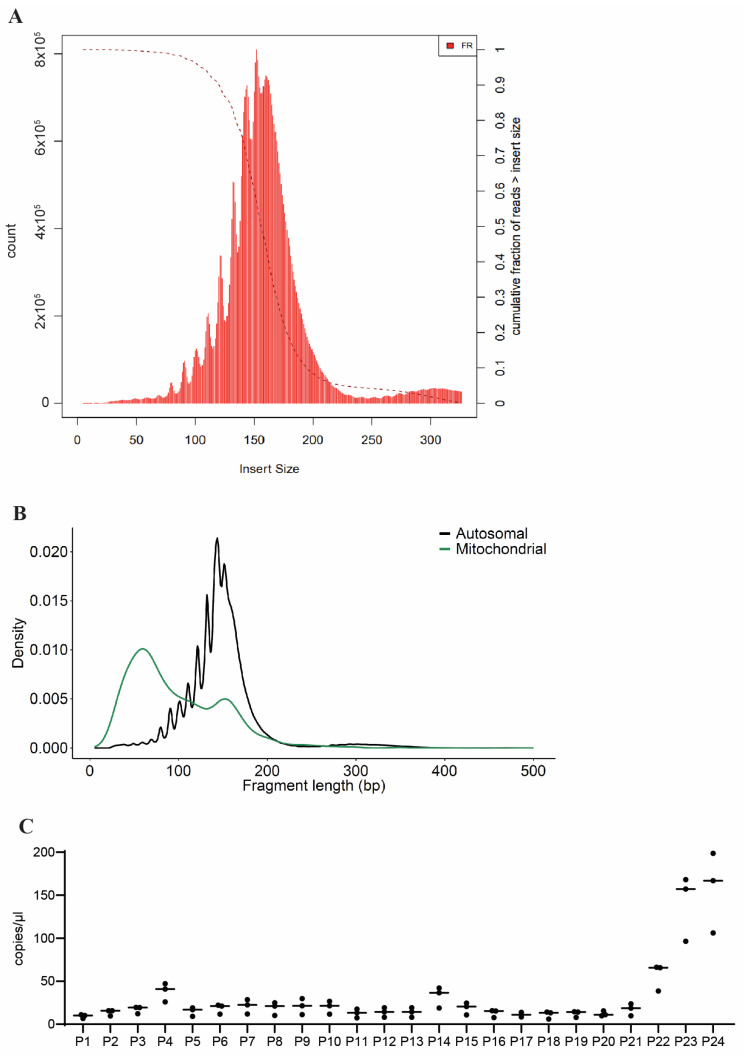
WGS reveals novel and consistently detectable sequences in cfDNA recovered from cerebral organoids. (**A**) Histogram showing the length of sequenced cfDNA reads recovered from mature cerebral organoids. (**B**) Length distributions of sequenced cfDNA mapping to either mitochondrial or autosomal origin. (**C**) Novel sequences detected in cfDNA from day 41 cerebral organoids using custom ddPCR probes (labeled P1–P24). Graphs show individual data points from n = 3 biological replicates and associated means.

**Figure 4 ijms-25-05522-f004:**
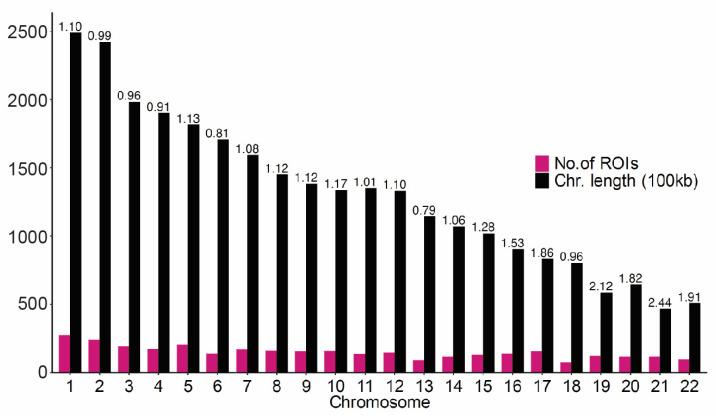
Distribution of ROIs across chromosomes. Bar plot displays the number of ROIs per chromosome (pink) and chromosome length in 100 kb (black) on the *y*-axis for each chromosome, with the ratio of ROI counts to chromosome length indicated above each bar.

**Figure 5 ijms-25-05522-f005:**
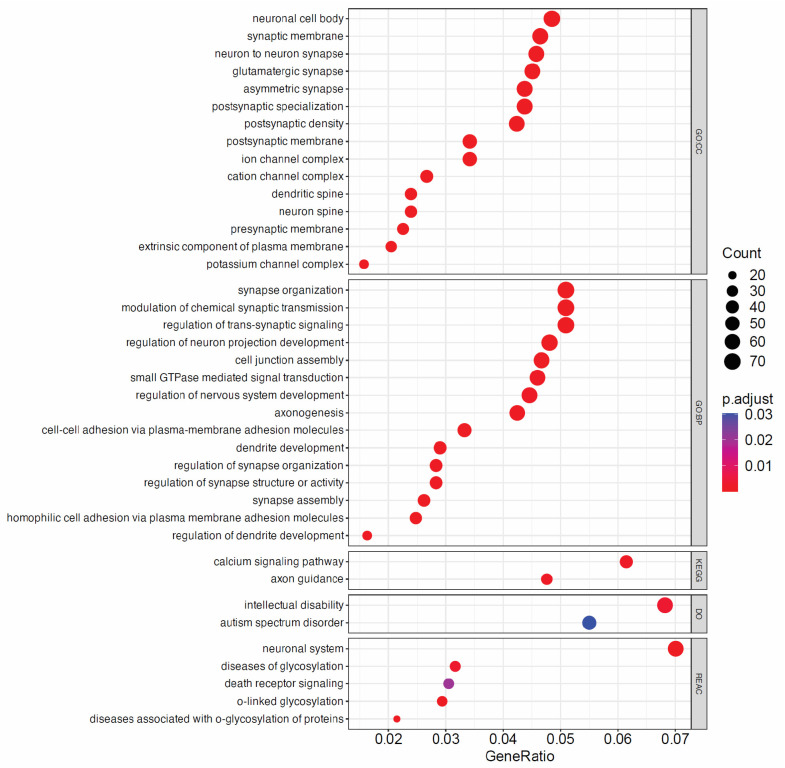
Gene Ontology (GO) classification of genes aligning with sequenced cfDNA ROIs. Shown are the top 15 terms from GO—Cellular Component (CC), the top 15 terms from GO—Biological Pathway (BP), terms identified using the KEGG and Disease Ontology (DO) databases, and the top 5 terms from the Reactome (REAC) database. GeneRatio is calculated as the number of input genes/total number of genes belonging to the source gene set, while p.adjust indicates the Benjamini–Hochberg-adjusted *p*-value for the over representation test, which is based on the hypergeometric distribution.

**Figure 6 ijms-25-05522-f006:**
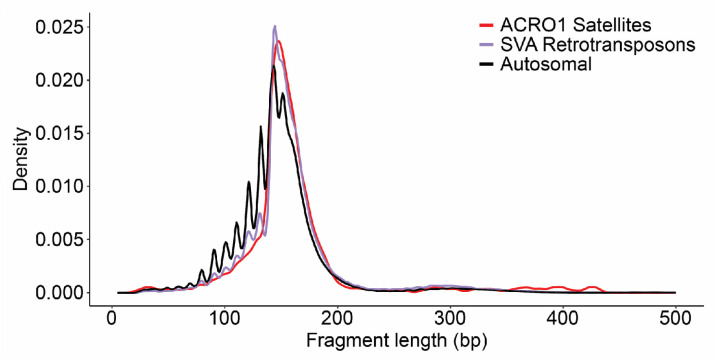
CfDNA aligning with retrotransposons and acro satellite repeats shows distinct fragment length distributions. Shown are length distributions of sequenced cfDNA ROIs intersecting retrotransposons or acro satellite repeats.

## Data Availability

The data presented in this study are available on request from the corresponding author.

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
