# Peer review of "Isolation and Characterization of Cell-Free DNA from Cerebral Organoids"

_ijms, 2024, doi:10.3390/ijms25105522_

Round 1

Reviewer 1 Report

Comments and Suggestions for Authors

Cell-free DNA (cfDNA) could be used as versatile disease markers, particularly for neurological disease, because monitoring brain tissue directly through biopsies is invasive and high-risk. Therefore, this research is very interesting and important for the early and timely diagnosis and treatment of neurological diseases. The authors obtained cerebral organoids from human embryonic stem cells, isolated cfDNA from these organoids both mitochondrial and genomic origin, then characterized the cfDNA and carried out sequencing as well bioinformation analysis. Overall, this manuscript is well organized. Some minor issues needs to be improved before it's accepted for publication.

Check "4. Materials and Methods" throughout and carefully. For example:

1. Line 325: 37oC and 5% CO2

2. Cultures were maintained for no more than 10 passages, provided no  morphological changes indicative of differentiation were present.

Comments on the Quality of English Language

Quality of English expression is good. But the symbols, formlas, expressions, etc. need to be checked very carefully.

Reviewer 2 Report

Comments and Suggestions for Authors

In this manuscript, the authors explore the possibility of obtaining cell-free DNA (cfDNA) from human cerebral organoids. Collecting cfDNA has shown significant promise for diagnosing health issues in pregnant mothers & fetuses, cancer patients and individuals with certain neurologoical diseases. Recent work (including from this lab) has demonstrated success in obtaining cfDNA from several types of organoids, but this approach in cerebral organoids has not been performed. The authors demonstrate that cfDNA can be obtained from cerebral organoids and quantified, demonstrating enrichment for genes associated with neurodevelopment. Their careful analysis also identified contamination of samples with rodent DNA, likely from the Matrigel, which highlights the care researchers must take to ensure the cfDNA they observer is from the expected source. Overall the manuscript was well-written and the conclusions largely followed from the experiments. My specific comments are below:

1.        I applaud the authors for noticing that the cfDNA is contaminated by rodent DNA, especially at the early timepoints. I think the Matrigel is a likely culprit, although as best I can tell the Corning Matrigel is made from mouse sarcoma cells (not rat). So it’s unclear where the rat DNA came from. Did the authors every perform WGS from early-stage organoids prior to Day 7 (introduction of Matrigel)? This would be the best evidence that the rodent DNA did in fact arise from the Matrigel and not other potential contamination sources. If not, could the authors perform qPCR or another assay to confirm the presence of rodent DNA arises specifically when Matrigel is introduced to the embryoid bodies/organoids?

2.        Relatedly, how does this observation affect the cfDNA results presented in Figure 2A-B. The remaining data was obtained from Day 41 organoids to avoid this issue, but the rodent DNA is the primary source of cfDNA at Day 10 (based on Sup Fig 1, and it’s not even close). Doesn’t that mean that the data at D10 in Fig 2A-B is severely compromised, and in fact the ‘outlier’ of the normalized data in Fig 2B is extremely skewed by this large presence of mouse DNA in the cfDNA sample? I’m not sure how the authors should best deal with this, but personally I think they should remove the timecourse data in Fig 2A-B because several of the timepoints contain high amounts of rodent DNA and thus analysis at this timepoints is suspect (through no fault of the authors). They could start this section discussing the rodent DNA contamination and state that all data was taken from Day 41, and just remove the timecourse cfDNA because I have significant issue with interpreting the Day 10-21 data. 

3.        Figure 2A: The figure legend states ‘Data points represent…’. But there are no data points in the graph, only the average with sem. The authors should either add the data points into the graph (as they did in D-E), or remove this statement.

4.        In lines 256-262 in the Discussion, the authors introduce new analysis of their data: looking at DMR regions in their dataset. I don’t understand why this analysis is in the Discussion, it adds additional insights into their dataset. It should be bumped up to the Results section, as most journals to not permit new analysis in the discussion.

5.        This inquiry highlights my naivete when it comes to cfDNA. Based on the introduction and my brief search, it seems like the most likely source of cfDNA is dead/dying cells spewing DNA into the environment. But I was wondering if it comes solely from genomic DNA (gDNA), or if cfDNA contains cDNA as well? In the discussion (lines 249-253), the authors cite several references about how epigenetic changes and altered methylation can alter the composition of cfDNA. So I assume this is evidence that more open/accessible regions are biased to be found in cfDNA, whereas genomic loci that are inaccessible due to H3K27me (or other marks) are less likely to be present in cfDNA. But if the cfDNA also contains cDNA (and thus actively transcribed genes), then that could easily explain the biased presence of neuronal genes in cfDNA. Is cDNA a feasible source of cfDNA, or is it definitively from gDNA?
